# Logically-Centralized SDN-Based NDN Strategies for Wireless Mesh Smart-City Networks [†]

**Sarantis Kalafatidis** [1,*] , **Sotiris Skaperas** [1] , **Vassilis Demiroglou** [2] , **Lefteris Mamatas** [1,*]
and **Vassilis Tsaoussidis** [2]

1   Department of Applied Informatics, University of Macedonia, 54636 Thessaloniki, Greece
2   Department of Electrical and Computer Engineering, Democritus University of Thrace, 67100 Xanthi, Greece
*   Correspondence: kalafatidis@uom.edu.gr (S.K.); emamatas@uom.edu.gr (L.M.)
†   This paper is an extension of our paper "Experimenting with an SDN-Based NDN Deployment over Wireless Mesh Networks" published in the IEEE INFOCOM 2022—IEEE Conference on Computer Communications Workshops (INFOCOM WKSHPS), Virtual, 2–5 May 2022.

**Abstract:** The Internet of Things (IoT) is a key technology for smart community networks, such as smart-city environments, and its evolution calls for stringent performance requirements (e.g., low delay) to support efficient communication among a wide range of objects, including people, sensors, vehicles, etc. At the same time, these ecosystems usually adopt wireless mesh technology to extend their communication range in large-scale IoT deployments. However, due to the high range of coverage, the smart-city WMNs may face different network challenges according to the network characteristic, for example, (i) areas that include a significant number of wireless nodes or (ii) areas with frequent dynamic changes such as link failures due to unstable topologies. Named-Data Networking (NDN) can enhance WMNs to meet such IoT requirements, thanks to the content naming scheme and in-network caching, but it necessitates adaptability to the challenging conditions of WMNs. In this work, we aim at efficient end-to-end NDN communication in terms of performance (i.e., delay), performing extended experimentation over a real WMN, evaluating and discussing the benefits provided by two SDN-based NDN strategies: (1) a dynamic SDN-based solution that integrates the NDN operation with the routing decisions of a WMN routing protocol; (2) a static one which based on SDN-based clustering and real WMN performance measurements. Our key contributions include (i) the implementation of two types of NDN path selection strategies; (ii) experimentation and data collection over the w-iLab.t Fed4FIRE+ testbed with real WMN conditions; (ii) real measurements released as open-data, related to the performance of the wireless links in terms of RSSI, delay, and packet loss among the wireless nodes of the corresponding testbed.

**Keywords:** software-defined networks; wireless mesh networks; information-centric networking; named data networking; smart-cities





## 1. Introduction

Smart-cities networks are characterized by a large number of heterogeneous end-user devices with spatial diversity, deployed over city-wide network regions. In this framework, the smart-city ecosystems adopt the Internet of Things (IoT) technology to utilize the smart-city applications (e.g., e-health [1] and environmental quality notification [2]), which are typically associated with critical performance requirements, e.g., low delay, reduced communication overhead, and high resilience. Wireless technology appeared as a key feature to meet the aforementioned requirements, increasing the communication range of distant infrastructure-free IoT deployments. As a consequence, various wireless communication technologies (such as WiMax, LTE, 5G, and beyond) have been proposed to support communication in smart city environments, each of them bringing different advantages to the network [2,3].

Wireless Mesh Networking (WMN) seems to attract more attention in several case studies of smart cities (e.g., Stratford [2], CityLab [4], SmartShatader [5]) as is capable to support a wide range of wireless smart community areas in a multi-hop manner, with ease-deployable and low-cost infrastructure, while at the same time guarantees robust and efficient connectivity in dynamic environments (e.g., mobile users support). In summary, some of the advantages provided by WMN in smart cities are (i) flexibility supporting efficient interconnection of multiple distributed end-user devices; (ii) unstructured self-sustaining and self-configuring topologies, and (iii) integration of different networking technologies such as wired, optic-fiber, cellular and sensor networks [3]. Moreover, the multi-hop communication over the WMN can work as a complementary solution to other wireless networking technologies, e.g., to provide connectivity in areas where 5G coverage is not sufficient or achieve low delay in case the 5G RAN is far. However, the WMN backbone network may incorporate regional communication characteristics, such as areas with (i) static nodes, requiring efficient management of network resources [6]; (ii) mobile nodes, causing frequent instabilities in the network (e.g., topology rearrangements), and (iii) high signal interference [4].

On the other hand, Named-Data Networking (NDN) [7], an Information-Centric Networking (ICN) [8] architecture, has been proposed as a promising approach to match the IoT application requirements [9] in smart-city environments [10]. In particular, NDN architecture: (i) facilitates content retrieval from the network, as NDN packets contain data names instead of IP addresses and (ii) contributes to reduced communication overhead thanks to the in-network caching [11]. A typical example of NDN implementation on top of smart-city environments is the work [6] in which the authors examine the efficiency and effectiveness of NDN principles in CityLab testbed, using containerized NDN service placement in wireless nodes.

However, deploying NDN in such dense WMN requires employing additional mechanisms to enable efficient multi-hop NDN communication. On the one hand, an important aspect is the selection of nodes, in which, the NDN protocol will be deployed [12], to avoid unnecessary resource consumption. For example, Figure 1 depicts that in a multi-hop communication between two NDN nodes (producer-consumer), not all of the available intermediate nodes are required. On the other hand, a second major challenge is to adjust the NDN paths appropriately.

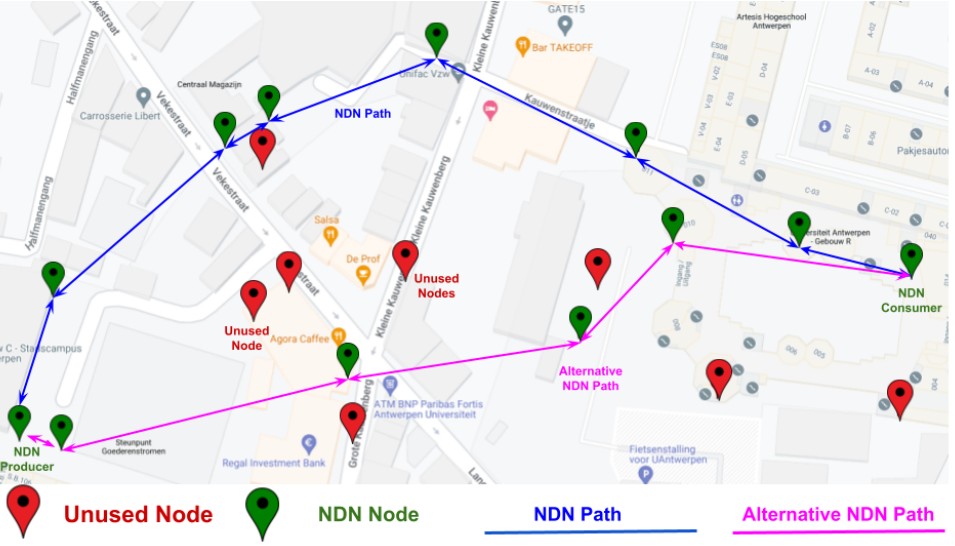

**Figure 1.** Example of NDN communication over CityLab testbed [4].

Considering the latter, smart-city networks provide a diverse set of communication conditions, spanning from stable to unstable conditions. For instance, a specific region of a smart-city deployment may suffer from unstable wireless conditions, such as frequent

topology rearrangements, wireless node failures, or high signal interference in crowded areas [4], affecting the NDN performance. In this context, efficient NDN packet forwarding over WMN networks with unstable conditions and low-quality wireless links is remaining an open issue [12].

Here, we argue that Software-Defined Network (SDN) may provide the missing features of intelligent centralized control and programmability to facilitate efficient NDN operation in challenging communication environments, such as the smart-cities WMN [2,13]. For example, in decentralized (e.g., non-SDN) NDN routing solutions, each NDN router is only aware of the state of its own paths and ignores other paths in the network. The SDN-NDN integration enables the centralized network view making easier the path selection/finding considering more sophisticated forwarding decisions based on several metrics (e.g., network state, RTT, hop counts, cash contents) [14]. Furthermore, the state of NDN routing nodes is maintained in low complexity as the routing decisions, and the network configuration functions are taken over by the SDN controller. Thus, the integration of SDN with NDN over WMNs may bring the following features: (i) global view of the network topology and centralized monitoring data collection; (ii) allows the decision-making (including NDN best-path selection) based on the "global" network view; and (iii) the dynamic NDN configuration in groups of network nodes. In this context, in our previous work [15], we proposed an SDN controller which utilized a WMN routing protocol, providing a reactive solution for NDN path selection.

On the other hand, the centralized control of SDN brings communication overhead which is an important tradeoff between the cost and the benefits of designing SDN forwarding strategies [14]. Nevertheless, there are several studies that target reducing the SDN overhead, especially over low-power SDN wireless networks, e.g., [16]. Additionally, a reactive solution, such as [15], may handle dynamic changes in the communication performance, this comes with the cost of increased management overhead, due to the frequent (and potentially unnecessary) NDN path changes. The latter implies that this approach could be inefficient over network typologies consisting of static wireless nodes [4]. Motivated by this fact, we discuss the limitations of a one-fits-all NDN path selection strategy, claiming that an appropriate solution should take into consideration the respective network environment, even in the context of a smart-city environment.

In this work, we discuss the trade-offs between two logically-centralized NDN strategies over WMN smart-city networks. More precisely, we consider the following SDN-based NDN path selection solutions, which they target efficient end-to-end NDN communication in terms of performance (i.e., delay): (i) a *reactive approach*, which is an extension of our previous work in [15] (this approach is utilized also in our work [17]) and re-adjusts the NDN path based on the BATMAN protocol; and (ii) a *proactive approach*, that a priori defines the NDN path, based on a combination of partitional clustering and similarity-based measures.

The key contributions of the specific work are summarized as follows:

- Implementation of an SDN-based system that supports two NDN path selection strategies (i.e., a reactive one and a cluster-based proactive), built on top of real WMN networks.
- Real experimentation over the w-iLab.t testbed [18], evaluating the performance impact of our SDN-based solutions considering experimental scenarios with stable and unstable conditions.
- Collection of real measurements related to RSSI, Delay, and Packet loss from indoor [18] WMN network. We record the performance of links among real wireless nodes and provide the measurements as released open-data [19].

The remainder of this paper is organized as follows. The next section contrasts our proposal against the related works. In Section 3, we detail the characteristics and design choices of our approach, providing an extensive experimentation analysis in Section 4. Finally, our conclusions and directions for future work are presented in Section 5.

## 2. Related Work

The smart community networks may benefit from the ICN approach as it can increase the network performance, reducing communication overhead, thanks to the in-network caching capabilities, as shown in [6]. However, ICN architectures, including NDN, lack of inherent mechanisms for supporting efficient NDN operation in challenging communication environments, such as WMN. In this context, content delivery and packet forwarding over such networks, with unstable wireless links, remains an open issue [12]. In this work, we study two different strategies for appropriate NDN path selection over WMN environments: (i) a dynamic approach with a reactive operation and (ii) a proactive one based on clustering evaluation of the quality of the wireless links.

Here, we discuss a number of NDN-based solutions sharing similar design characteristics with the aforementioned strategies. Moreover, we divide the representative-related works into dynamic/reactive and cluster-based approaches, comparing them with the basic technical characteristics of our strategies (i.e., SDN-based centralized control and real experimentation over WMN), as illustrated in Table 1.

**Table 1.** Related Work Comparison.

| Approach | Works | Centralized | SDN | Wireless Mesh | Real Experimentation |
|---|---|---|---|---|---|
| *Reactive* | SRSC [20] | ✓ | ✓ | X | X |
| | Multipath Forwarding [21] | ✓ | ✓ | X | ✓ |
| | Software-Defined NDN [22] | ✓ | ✓ | X | X |
| | SDN-NDN over WMN [23] | ✓ | ✓ | ✓ | ✓ |
| | **Our reactive strategy** | ✓ | ✓ | ✓ | ✓ |
| *Cluster-based* | Cluster-based NDN Routing [24] | X | X | X | X |
| | LCRN [25] | X | X | X | X |
| | NDN-based IoT [26] | X | X | X | X |
| | PiGeon [6] | ✓ | X | ✓ | ✓ |
| | **Our proactive strategy** | ✓ | ✓ | ✓ | ✓ |

Recent approaches have been trying to resolve NDN routing and forwarding limitations with SDN [27]. In [20], authors designed and evaluated an SDN-based routing scheme for CCN/NDN (SRSC) which fully exploits the NDN principles, and, thus, the Controller and the nodes communicate using NDN messages (exchanging control and information messages). Particularly, the controller makes the routing decisions, and the NDN nodes act as forwarding devices only, as in our case. In SRSC, the Controller only informs an NDN node of the entire path to the content and afterward, the NDN nodes communicate with each other (hop-by-hop), to create the specific path. In contrast, we selected the controller to communicate independently with each on-path node and configure the NDN network, because of the unstable mesh topology.

In [21], the authors proposed an SDN-enabled Controller for multipath forwarding in NDN. The SDN controller analyzes the global view of the NDN network and makes appropriate forwarding decisions, according to the router states, the available forwarding paths, and the cached contents. The particular centralized solution improves the performance of NDN compared to a distributed multipath forwarding strategy, which relies on *a priori* forwarding information and is inappropriate for networks with dynamic topologies, as in our case. Such solutions have been evaluated over a real-world WAN network, so frequent path changes and unstable topologies were not investigated in depth.

Authors in [22] introduced an integrated SDN-NDN framework and modified NDN's FIB design to address FIB overflow. In particular, FIB overflow is affected by (i) a large number of different contents and (ii) long-lived FIB entries. In our work, we maintain the default NDN's FIB design and address those issues by creating short-lived FIB entries in the NDN network.

In [23], authors target to improve the NDN performance through efficient content management, considering features of wireless communication. Their proposed solution

has been evaluated in a real WMN environment and exhibits the most common design features with our dynamic approach (e.g., SDN-based centralized control over real WMN). Nevertheless, the main difference between our approach with [23] is decision-making regarding the dynamic conditions of WMN. In particular, our work aligns the functionality of NDN with the decisions of a dynamic WMN routing protocol, offering a rapid response of the NDN network to frequent topology changes (e.g., due to link failures).

Most of the aforementioned works are not validated in real-world test beds. Moreover, there is no prior work that addresses the challenging communication issues of deploying NDN in real-world mesh networks, using the SDN approach, as illustrated in Table 1.

As our proactive strategy, there are also other works that utilize clustering methods for NDN routing. In [24], authors propose an NDN routing protocol for wireless networks using clustering to reduce the number of nodes considered for route discovery. Moreover, in [25], a routing protocol is presented based on clustering to reduce routing overhead among the network nodes. In [26], the authors studied the integration of NDN-based IoT with Edge computing, introducing an algorithm that employs clustering to improve NDN routing. In contrast with our cluster-based approach, the above works are not evaluated over real network environments, and their routing decisions do not take into account WMN network characteristics (e.g., the quality of the wireless links).

In work, [6] authors use clustering for NDN service placement and investigate ICN advances by conducting experiments over a smart-city WMN test-bed [4]. The decision-making for the service placement takes into account compute resources (i.e., CPU) and bandwidth among the network nodes. In contrast to this work, we investigate the best NDN path selection considering RSSI and delay metrics among the network nodes, as we target efficient NDN communication over WMN in terms of delay and reliability.

From the literature review, we observe that most cluster-based routing works (as the above) mainly focus on excluding monitoring data or characteristics of the network (e.g., considered network nodes) that affect the routing decision, aiming to reduce the data communication latency or costs. Compared to these works, we approach cluster-based routing differently, using a multi-cluster approach as an evaluation quality measure to select the best NDN path (i.e., the clustering results are taken into account for the path selection). The employment of a clustering-based approach, which intuitively requires stationary data structures, corresponds with the results of an extensive experimental analysis over a real wireless network with static nodes (described in Section 4).

## 3. Proposed SDN-Based System

In this section, we present the proposed SDN system and the implemented reactive and proactive approaches, which target efficient NDN operation over WMNs. The main objective of both approaches is the selection of the appropriate paths between the NDN producer and consumer. In summary, we consider:

1. A reactive NDN path selection strategy that aligns NDN paths with the dynamic routing decisions of the WMN protocol. This approach is based on distributed decision-making information (i.e., each node chooses the best route for each destination among its neighbors) which is collected from the SDN Controller to configure the NDN nodes.
2. A proactive NDN path selection strategy based on evaluating available wireless links in terms of RSSI and delay. Specifically, in this approach, we collect historical network monitoring data and classify their performance to select the appropriate NDN path.

Their detailed description follows next.

### 3.1. Reactive NDN Path Selection Strategy

Here, we present our proposed SDN-based system and its corresponding mechanisms, aiming to the flexible and adaptive NDN operation in unstable WMN regions of smart-city backbone networks. Its main functionalities are (i) centralized monitoring of the wireless mesh network; (ii) dynamic best-path decision-making, and (iii) NDN configuration according to the selected routes. The system consists of two functional entities: (i) *the SDN*

*Controller* which is the centralized control point of the network and (ii) *the Network Nodes*, which support NDN communication over WMN. A detailed description of the system components follows.

### 3.1.1. SDN Controller

The Controller is the key component that enables the integration of the NDN with the dynamic decisions of wireless routing protocol [28] and as it performs centralized monitoring of the wireless nodes to configure the NDN paths. Its main functionalities are (i) information collection about the network state and rapid detection of network changes, enabling the global view of the entire network in real-time; (ii) definition of the best route for each content request between the NDN-Consumer and the NDN Producer, and (iii) to NDN route establishment including per-hop NDN face and FIB entry creation. The overview of the Controller's operation is presented in Figure 2.

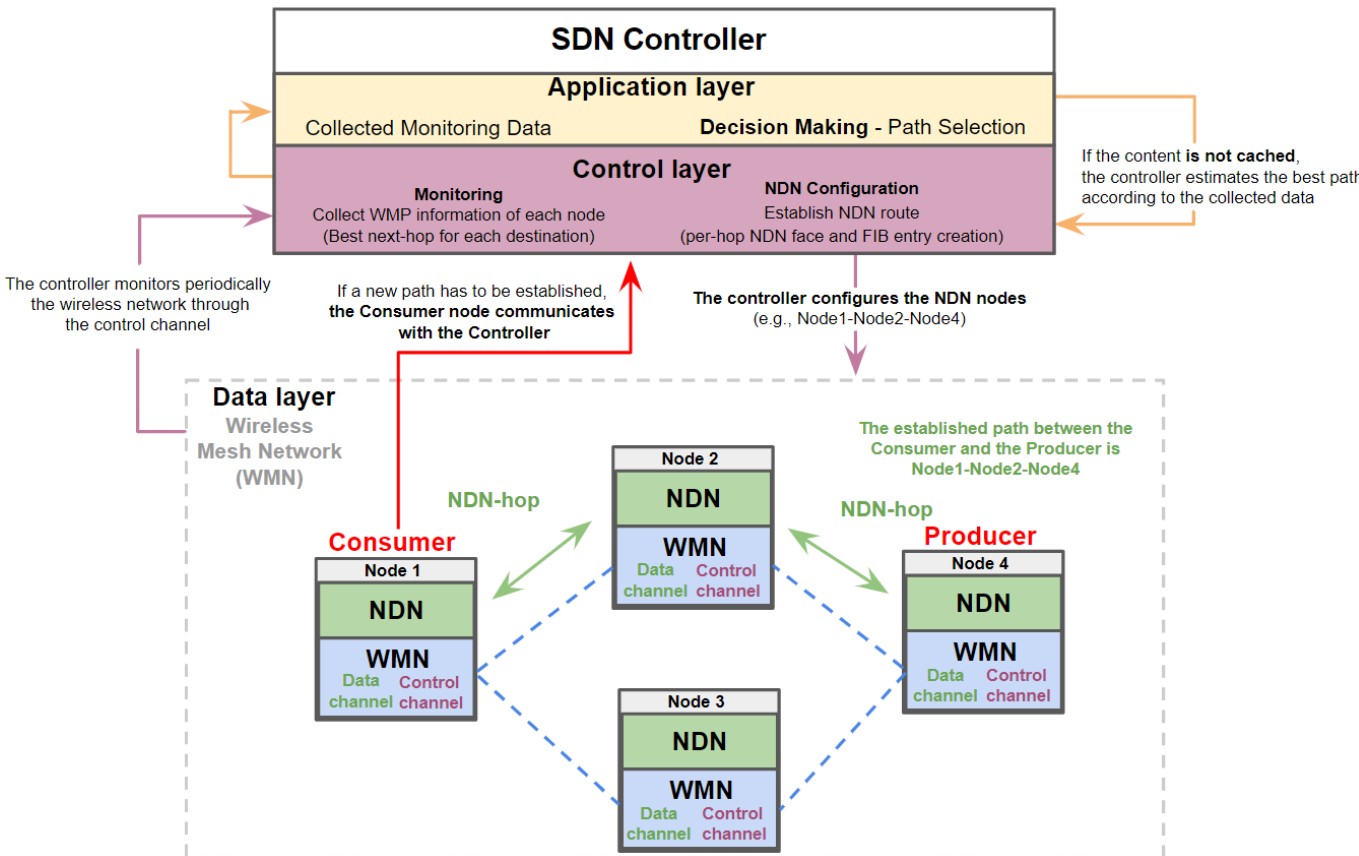

**Figure 2.** SDN-based Experimentation System.

The monitoring data collection of WMNs is performed through the WMNs routing protocol, i.e., the blue dashed lines of Figure 2. That is the centralized information collection from the distributed wireless nodes, regarding the discovery of the neighbors of each node and the routes that occur between the hops of the network. The routing costs among the links are based on the quality metrics of BATMAN [28] routing protocol (Layer 2), which is utilized in our implementation. The Controller is also located in the WMN network and communicates with the Network Nodes over IP.

The Controller manages the content requests through the reactive operation, following these four actions: (i) receives Consumer's updates for each new Interest packet; (ii) defines the best wireless routing path among the Consumer and the Producer, according to the collected monitoring data; (iii) establishes the selected NDN route, and (iv) triggers the Consumer to send the particular Interest packet.

Moreover, our system maintains information related to NDN in-network cashing. Especially, our Controller associates the content prefix with the corresponding utilized NDN paths and the content's Freshness Period to determine whether the content is cached. More precisely, as the Data packet will travel through the reverse path, the Controller node stores the path that has cached the particular Data packet and estimates the caching remaining time. This can be accomplished by maintaining information about the particular data packet (e.g., Freshness Period) as well as the corresponding caching information (e.g., utilized cache size, remaining entries, caching policy).

The Controller is located in the wireless mesh network and communicates with the wireless nodes over wireless links. Here, a major challenge is to guarantee reliable communication between the SDN controller and the WMN nodes, for example, in case of disruptive communication. Following our previous works [29,30], we assume two wireless communication channels: (i) a control channel utilizing long-range but low-data-rate wireless communication, targeting the reliable connection between the Controller and the wireless nodes and (ii) a data channel utilizing a high-data-rate and short-range channel, supporting application data transfer. Although, the deployment of a reliable control channel is a complex aspect, and deserves an independent study outside the scope of this paper. We briefly discuss this approach here, to underline the complexity of the above task, which may become a major disadvantage considering the reactive strategies.

### 3.1.2. Network Nodes

The infrastructure of our proposed system consists of interconnected wireless nodes that support NDN communication. The NDN and WMN functionalities are independent and are integrated with the Controller (i.e., the Controller monitors the WMN to configure the NDN). Here, we describe the *NDN and WMN functionalities* to illustrate the system's network nodes operation.

In a nutshell, NDN is a future Internet architecture that follows the ICN principles and accomplishes named content retrieval by employing two types of packets (e.g., the Interest and the Data packets). In NDN, Consumers send an Interest packet in the network to fetch the corresponding Data packet that contains the requested content. Although a Data packet is originally generated from a Producer node, it may be retrieved from intermediate nodes' caches, as NDN supports in-network caching.

Each NDN node uses three components: the Content Store (CS), the Pending Interest Table (PIT), and the Forwarding Information Base (FIB). When a new Interest packet is received, the NDN node first checks if the requested content exists in the CS (i.e., it is cached), and in that case, it responds directly with the Data packet. Alternatively, if the prefix matches a specific PIT entry (which contains the already sent Interest prefixes associated with the respective faces), then the incoming face is added to the particular entry (meaning that when the Data is fetched it will be forwarded also to that face). Otherwise, a new PIT entry is created and the Interest is forwarded to the next hop according to the FIB information [11].

In our NDN deployment, face creation and prefix registration are triggered from the Controller node. Since NDN communication is Consumer-driven, our system performance heavily relies on the Consumer node behavior. Thus, this plane targets to fetch the data efficiently with the minimum communication delay, by exploiting the NDN features.

The NDN consumer communicates with the SDN controller, if a new Consumer-Producer path has to be established, i.e., in cases: (i) the consumer requests a specific content for the first time or (ii) freshness period for a content (already requested) has expired. In all other cases, the requested content is retrieved to the consumer from the network, using the in-network cashing capability of NDN.

Here, we give an example of the NDN and SDN interaction of our system, as illustrated in Figure 2. The NDN Consumer (node1) intends to fetch the content with the prefix *e.g.*, *sensor/temperature*. Thus, it sends a request to the Controller to inform them about the specific content. If the content is not cached in the network, the Controller finds the best

path to the NDN Producer (e.g., Node1-Node2-Node4) and establishes the NDN routes. Finally, the Controller triggers the Consumer to transmit the Interest packet through the created path.

### 3.2. Proactive NDN Path Selection Strategy

The proactive strategy is also SDN-based and it is grounded on the implementation described in Figure 2. Specifically, each node in the network collects information from its neighbors related to RSSI and delays, periodically. The collected monitoring data is transferred to the SDN controller through the control channel. Then, the controller performs the clustering and chooses the best path by adjusting the NDN on the wireless nodes accordingly. Compared to the reactive strategy, in this implementation, the SDN controller forwards all the content requests on the selected path until it re-estimates the best path again. For this reason, we consider this strategy proactive.

More precisely, the proactive strategy is based on partitional clustering and similarity-based distance measures. The main goal is to determine the best NDN path in terms of performance and reliability for end-to-end NDN communication (e.g., between an NDN consumer and NDN producer). In summary, our implementation has the following objectives: (i) the data collection among the network nodes, regarding the RSSI and delay, in a central node of the network; (ii) the clustering of the measurements, and (iii) the determination of the end-to-end NDN best paths among the wireless links.

In Figure 3, we illustrate the NDN path selection according to the clustering results, considering 4 clusters, sorted from best to worst based on the average value within the cluster. The color gradations indicate the cluster each link belongs to, i.e., the colors green, blue, black, and red symbolize the clusters from best to worst, respectively. Then, each provided path is characterized by its worst link, e.g., the worst path contains at least one link clustered as the worst link. For example, in Figure 3, the best path is that containing the nodes 1-2-3-5-8, since all the including links are clustered as best-links (notated with green color).

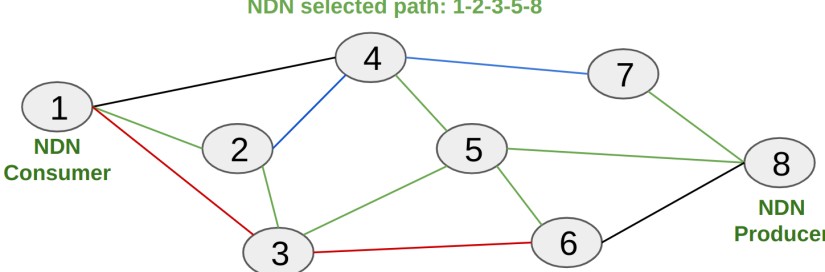

**Figure 3.** NDN path selection using the proactive strategy.

Here we describe the clustering method, we choose the dynamic time warping (DTW) [31] to estimate the similarity between the time series. Exploiting the nonlinear and time-independent nature of DTW, compared to typical measures (e.g., Euclidean distance), which provides a more robust and "global" (less sensitive to local time-series shifts) similarity estimation, improving the clustering performance. Moreover, we utilize the k-medoids [32] algorithm, since it is a representative object technique, more resilient to outliers [33] than k-means. The fundamental steps of the proposed clustering approach, are given below:

- Step 1: Calculate DTW, pairwisely, between all the available links.
- Step 2: Apply the k-medoids algorithm to the vector that describes the pairwise similarity of all the available links (computed in Step 2). The number of clusters is pre-defined (in Section 4. We chose 4 clusters).
- Step 3: Sort the clusters, according to a specific Quality of Service (QoS) metric, for example, RSSI or delay.

## 4. Experimental Evaluation

In this section, we assess our proactive and reactive approaches for NDN path selection, targeting to the efficient operation of NDN over WMN. Our evaluation includes real experimentation over an indoor wireless environment, considering small-scale SDN-NDN deployments, utilizing the w-iLab.1 Fed4FIRE+ testbed. In summary, the evaluation metrics include (i) the NDN performance in terms of delay; (ii) the SDN communication management overhead; (iii) the frequency of dynamic path changes (path selection), and (iv) NDN reliability (i.e., percentage of Interest-Data exchange failures). More precisely, we consider two experimental scenarios:

- Scenario 1, which evaluates the reactive approach over different types of network conditions, i.e., stable and unstable WMN conditions.
- Scenario 2, which assesses the effectiveness of a proactive (clustering-based) approach in this context, also comparing its outputs with these of the reactive approach.

### 4.1. Experimentation Setup

Here, we present the most important experimentation details of both scenarios. The NDN implementation is based on the containerized Named data Forwarding Daemon (NFD) [34], developed in Docker containers [35] since it is lightweight, easily reconfigurable, and facilitates the NDN deployment on any hardware. Moreover, we use the better approach to mobile ad-hoc networking (BATMAN) [28] wireless mesh routing protocol.

The multi-hop network topologies consist of wireless network nodes (Intel NUC devices) of the w-iLab.1 testbed. The nodes are equipped with an AR9462 wireless network adapter that we used to construct our wifi mesh network. We used the ath9k driver and configured these devices to run at 2.4 GHz with 20 MHz channel width in mesh mode. Moreover, we use a separate 802.11 wireless channel for the control messages (i.e., for the communication of the controller with the NDN nodes,) which is in mesh mode and uses the ath10 driver.

Regarding the performance measurements of the wireless links (i.e., RSSI and delay) utilized in the clustering process, we use wireless connectivity (among all nodes of the network) based on a Peer Link Management protocol, which is used to discover neighboring nodes and keep track of them. Here, the neighbor discovery is only limited to the signal range of each node. The evaluation is based on *Ping* tool by sending a batch of 100 ICMP packets every 3 s. As illustrated in Figure 4, the results present: *total* completion time—the total time to communicate of the 100 packets (we use a 2-s threshold waiting for each batch to complete. If the time is exceeded the particular batch is considered undelivered); *PLR*—the percentage of ICMP packets loss (i.e., out of 100); *avg*—the average Round Trip Time (RTT) of the ICMP packets; *sd*—and the Standard deviation RTT of the ICMP packets. The *RSSI* measurements are recorded from a second interface on each wireless node (the first interface was in mesh mode and was used for sending the ICMP packets), which is in monitor mode and uses the same driver. The performance measurements were collected from the w-iLab.1 testbed [18], utilizing the above method, are provided in [19].

Finally, the use case under consideration is an IoT scenario, where an IoT NDN Consumer requests sensor measurements from an IoT NDN Producer, that generates emulated sensor measurements. Here, focus on two IoT applications with different requirements: (i) short flows with delay demands, assuming a typical sensor measurement application, i.e., the produced data packets have a limited size (350 Bytes) and represent raw sensor measurements (e.g., temperature, humidity, etc); and (ii) long flows with throughput demands, assuming photo file transfers with a limited size (1 Mb). In the short flows, the emulation of Consumer and Producer applications are based on the *ndnpeek and ndnpoke* tools, respectively, and for the long flows emulation, we use the *ndncatchunks and ndnputchunks* tools to transfer files as data segments. Moreover, to measure NDN connectivity failure attempts, we have disabled NDN retransmissions for the short-flow application.

```
RSSI    total            PLR              avg      sd
-78, time 752ms, 1.96078% packet loss, 9.110, 9.442 ms
-80, time 1985ms, 68.4564% packet loss, 7.761, 4.823 ms
-79, time 1097ms, 10.7143% packet loss, 15.263, 18.432 ms
-79, time 1999ms, 85.0394% packet loss, 31.592, 15.728 ms
-80, time 1996ms, 8.41121% packet loss, 151.568, 322.692 ms
-80, time 691ms, 3.84615% packet loss, 14.272, 23.896 ms
-80, time 1993ms, 38.0282% packet loss, 40.336, 89.875 ms
-78, time 1992ms, 33.6134% packet loss, 84.310, 117.945 ms
-78, time 1993ms, 86.5546% packet loss, 10.433, 8.622 ms
-78, time 1737ms, 28.5714% packet loss, 54.498, 57.614 ms
-78, time 1463ms, 11.5044% packet loss, 52.888, 54.362 ms
-78, time 1168ms, 11.5044% packet loss, 30.180, 42.028 ms
-78, time 1692ms, 20% packet loss, 27.511, 21.486 ms
-78, time 1990ms, 39.1608% packet loss, 73.352, 80.365 ms
-78, time 951ms, 3.84615% packet loss, 10.745, 7.691 ms
-78, time 1994ms, 58.9041% packet loss, 281.733, 151.220 ms
```

**Figure 4.** Evaluation of the wireless links.

### 4.2. Scenario 1—Evaluation of the Reactive Processes

Here, we demonstrate the functionality of the reactive strategy, targeting the flexibility and adaptability improvements that may bring, rendering NDN deployments over dynamic WMN topologies feasible.

In this scenario, we focus on the first IoT application scenario, i.e., short flows with delay demands. More precisely, we consider 150 different IoT contents generated from the Producer node, while the Consumer node transmits 1000 interest packets one by one to fetch the generated IoT contents from the Producer node. The content requests follow the Zipf distribution [36], (with $\alpha = 1.5$). We set the freshness period to 10 s, which is the time that the cached content is valid.

First, we discuss the results over stable networking conditions. In practice, we consider a multi-hop network topology consisting of seven wireless network nodes from the 9th floor of the w-iLab.1 Fed4FIRE+ [18] office lab, illustrated in Figure 5. The experiment assumes one Consumer (Node9-13), one Producer node (Node9-21), and the Controller (Node9-3) participating in the same wireless topology, i.e., the latter performs centralized control of the entire network. Although the selected devices are located in close proximity, we adjust the transmission power (TP) of the wireless nodes to 3 dBm and managed to shape a multi-hop scenario. The monitoring data of each node include information about the BATMAN neighbors and originators enabling the global view of the network, and, are sent to the Controller through the A.L.F.R.E.D. tool [37]. The Controller configures the NDN nodes by transferring messages to the NFD container over TCP. These messages include information about the next hop and the particular content prefix and, thus, enable NDN face creation and prefix registration. Moreover, the nodes are not located nearby, thus, their connectivity is accomplished through the WMN. Results are collected over 10 repetitions of the experiment. Furthermore, the evaluation includes three metrics: (i) the performance of the NDN network in terms of communication delay (i.e., the Interest-Data exchange procedure between the Consumer and the Producer); (ii) the path establishment delay, i.e., the elapsed time at which the Controller makes the best path decision and configures the NDN network, and (iii) the performance of SDN decision making over the WMN. A detailed description of the metrics used, follows.

- $RTT_{NDN}$ is obtained from the NDN Consumer node and represents the round trip time (RTT) between the interest packet transmission and the data packet reception,

$$RTT_{NDN} = RTT_{NDN_c} + RTT_{NDN_p},$$

  where $RTT_{NDN_c}$ corresponds to the RTT values in cache-hit cases and $RTT_{NDN_p}$ is the measured RTT in case of Producer-fetched content. Note that, the Consumer can fetch

the requested content either from the Producer node or from the on-path nodes, due to the in-network caching.

- *Total Delay:* denotes the elapsed time between the Consumer's request to the Controller until the Data packet reception, including the path establishment delay.
- *Best Path Changes (BPC):* represents the total number of best path changes in the whole duration of the experiment.

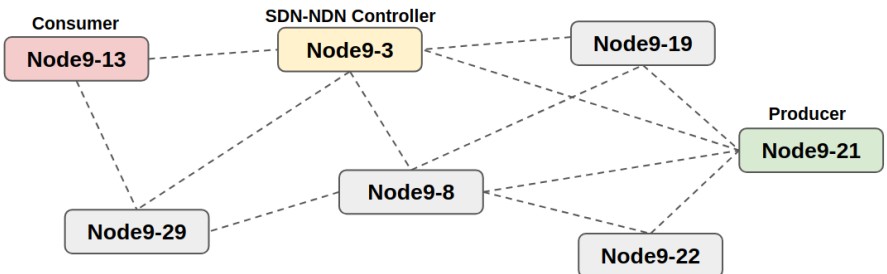

**Figure 5.** WMN topology, considering the 9th floor of the w-iLab.1 test.

We now move on to discussing our results. In Figure 6, we present the outputs of the reactive approach, considering the $RTT_{NDN}$ and the *Total Delay* metrics. The low average $RTT_{NDN}$ value, indicates that a reactive approach may ensure the seamless operation of the NDN in volatile network topologies, without compromising the NDN performance. On the other hand, the high average *Total Delay*, especially in contrast to the average $RTT_{NDN}$, reveals the additional network control overhead introduced by the Controller, as detailed in Section 3.1.1, which could be avoided in the case of proactive path selection strategies. Note that the significant difference between $RTT_{NDN_p}$ and $RTT_{NDN_c}$ ($\approx$87 msec) is attributed to the locally occurred cache hits in the Consumer node (without requiring any network transmission) while fetching data from the remote Producer requires interest and data packet transmissions over the 3-hop topology.

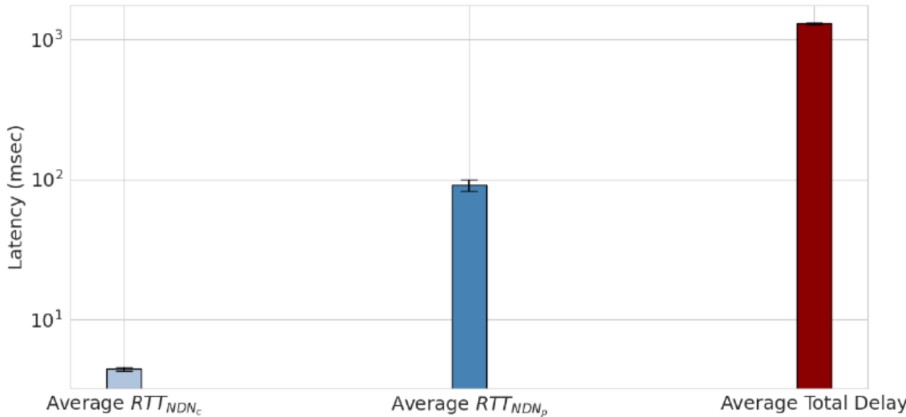

**Figure 6.** Average $RTT_{NDN_c}$, $RTT_{NDN_p}$ and *Total Delay*.

Figure 7 illustrates the average best paths hops and the total *Best Path Changes (BPC)*, per round (experiment repetition). As is shown, the average number of hops (in each round) is 3, confirming the validity of the experimental methodology. Additionally, the *Best Path Changes* deviation (i.e., ranging from 7 to 19) illustrates the wireless links volatility of the test-bed's setup and highlights the proposed system's capability to capture frequent network changes and establish the appropriate paths.

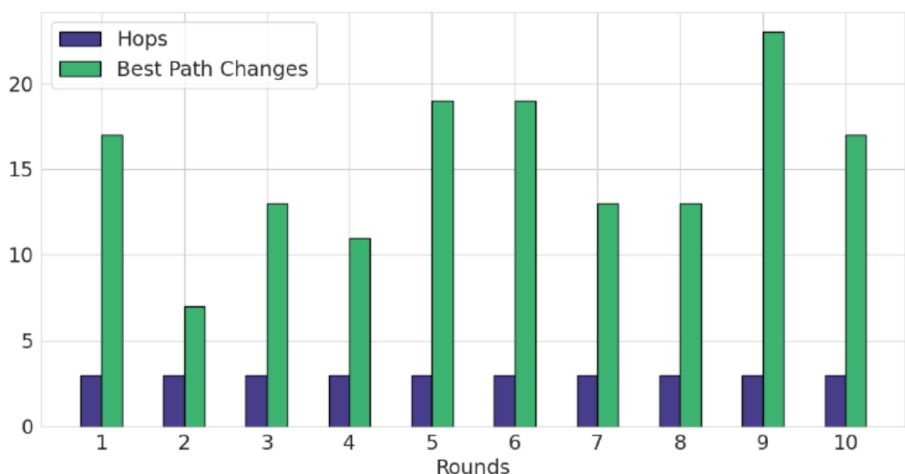

**Figure 7.** Number of Hops and *Best Path Changes (BPC)* pers round.

Subsequently, we discuss the outputs of the reactive process, targeting the effectiveness of NDN end-to-end communication, and, therefore, we exclude in-network caching (i.e., NDN cash contents). Here, the network topology consists of 6 network nodes from the 10th floor of the w-iLab.1 Fed4FIRE+ test-bed, as it is depicted in Figure 8. Node10-29 hosts the Consumer, node10-20 the Producer, and Node10-32 the Controller, which (as in the previous experiment) belongs to the same wireless topology. According to Figure 8, two multihop paths may be formed for the end-to-end communication, i.e., path1 (node10-29, node10-9 and node10-20) and path2 (node10-29, node10-32, node10-34, node10-5 and node10-20). The Transmission Power (TP) of node10-34 and node10-5 is 10 dBm, and the TP of node10-32, node10-9, and node10-29 is 5 dBm. Finally, we consider three types of communication conditions, by adjusting the Producer's TP, as follows: (i) 5 dBm (low TP), (ii) 10 dBm (high TP), and (iii) periodically increase/decrease from 5dBm to 10dBm, every 5 minutes (unstable TP).

Next, in Tables 2 and 3, we compare the proposed SDN, BATMAN-based, reactive solution with the conventional NDN approach, i.e., fixed path1 and path2 solutions, over the total amount of 1500 Interest-Data exchanges from the NDN Consumer node.

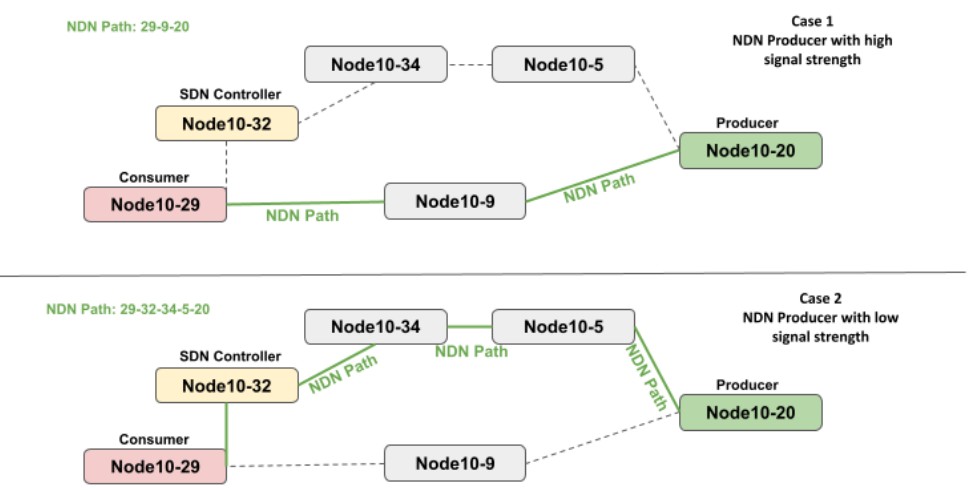

**Figure 8.** WMN topology, considering the 10th floor of the w-iLab.1 test-bed.

Table 2 enlists the average delay (msec) values of the Interest-Data exchanges for the fixed NDN paths and the reactive NDN solution. According to the outputs, Path2 provides the lower delay values for both low and unstable TP of the Producer, on the contrary, Path1 minimizes the delay for the case of high TP. Regarding the results of the proposed reactive strategy, the integration of the BATMAN protocol provides comparable performance in all

cases, implying that a dynamic approach can successfully select the best paths, according to the network conditions.

**Table 2.** Average of Interest-Data exchanges performance delay (msec).

| NDN Path | Low TP (5 dBm) | High TP (10 dBm) | Unstable TP (5–10 dBm) |
|---|---|---|---|
| Fixed Path1 | 61.62 | 29.86 | 53.71 |
| Fixed Path2 | 45.36 | 38.62 | 43.35 |
| Reactive solution | 48.58 | 29.93 | 48.99 |

Similarly, in Table 3, we evaluate the fixed and dynamic NDN path selection solutions in terms of reliability, i.e., comparing the total amount of Interest-Data exchange failures. The reactive strategy demonstrates its effectiveness in challenging communication conditions (low and unstable TP), providing the lowest number of interest-data failures, and also, indicating the potential gains of a dynamic mechanism, concerning unstable communication conditions. On the other hand, Path1 has the least failures in high TP cases.

**Table 3.** Total amount of failures over 1500 interest-data exchanges.

| NDN Path | Low TP (5 dBm) | High TP (10 dBm) | Unstable TP (5–10 dBm) |
|---|---|---|---|
| Fixed Path1 | 115 | 11 | 61 |
| Fixed Path2 | 70 | 53 | 58 |
| Reactive solution | 58 | 40 | 51 |

Finally, Table 4 presents the results of the reactive solution's path choices regarding path 1 and path 2 (i.e., how many times BATMAN chose each path), considering the different TP adjustments for the Producer node. As it is shown, the reactive solution's path choices are in accordance with the best path selection, derived in Tables 2 and 3. In other words, our dynamic approach, integrating the functionality of NDN with the decisions of a wireless mesh protocol (i.e., BATMAN), successfully detects the best paths.

In a conclusion, scenario 1 demonstrates that an SDN-based solution may effectively support the NDN operation over WMN, especially considering the reliable performance on challenging communication networks. However, real-time network monitoring increases the network overhead, also adding extra delays to the network performance due to the centralized SDN control management overhead. Hence, in scenario 2 we discuss the potential gains of a proactive routing strategy instead of the operation of dynamic NDN path changes, targeting relatively stable wireless networks.

**Table 4.** Reactive's solution path choices over 1500 interest-data exchanges.

| NDN Path | Low TP (5 dBm) | High TP (10 dBm) | Unstable TP (5–10 dBm) |
|---|---|---|---|
| Path 1 | 41 | 1186 | 362 |
| Path 2 | 1469 | 314 | 1138 |

*4.3. Scenario 2—Evaluation of the Proactive Strategy*

In scenario 2, we aim at the effectiveness of the SDN-based proactive procedure in terms of NDN performance over WMN smart-city environments. We use a multi-hop network topology consisting of ten wireless network nodes as illustrated in Figure 9 with green circles, including one Consumer (Node10-29) and one Producer node (Node10-20), located at the edges of the network.

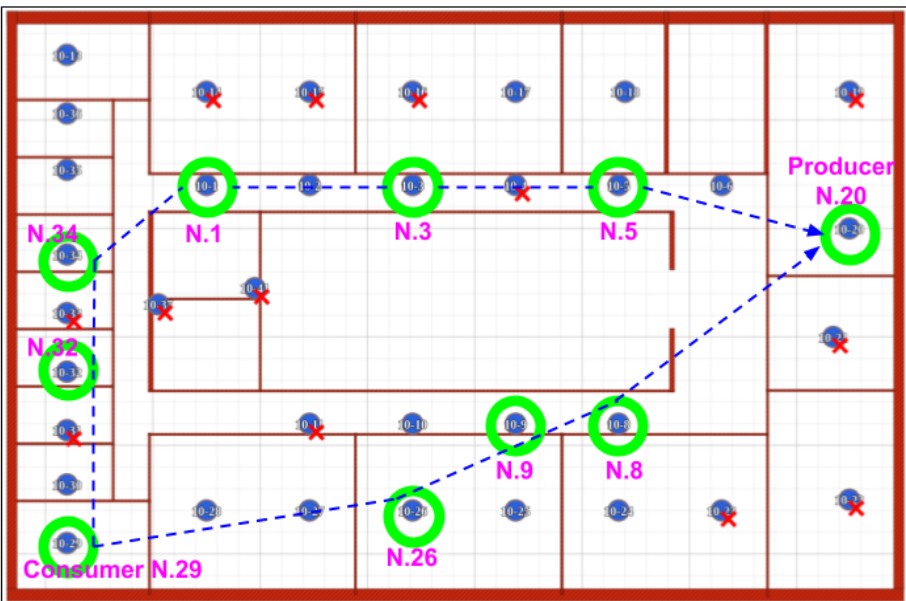

**Figure 9.** Selected topology—w-iLab.1 office lab 10th floor [38].

First, we discuss the clustering results over the experimental network topology. More precisely, the proposed clustering technique considers 4 clusters, classified from best to worst based on their intra-cluster mean value; the best cluster provides the minimum intra-cluster mean value. Figures 10–12 illustrate the clustering outputs considering as inputs the absolute RSSI, the delay, and the bivariate RSSI-delay values of the links, respectively. The color gradations denote the cluster that each link belongs to, i.e., green, blue, black, and red colors symbolize the clusters from the best to worst, respectively. According to the clustering results: (i) RSSI values mainly depict the distance among the nodes; as expected, since we experiment over an indoor environment with low interference, (ii) no evidence for the high similarity between RSSI and delay values exists, e.g., the path from node-1 to node-20, (iii) RSSI-delay-based clustering presents dissimilarities from both RSSI and Delay clustering.

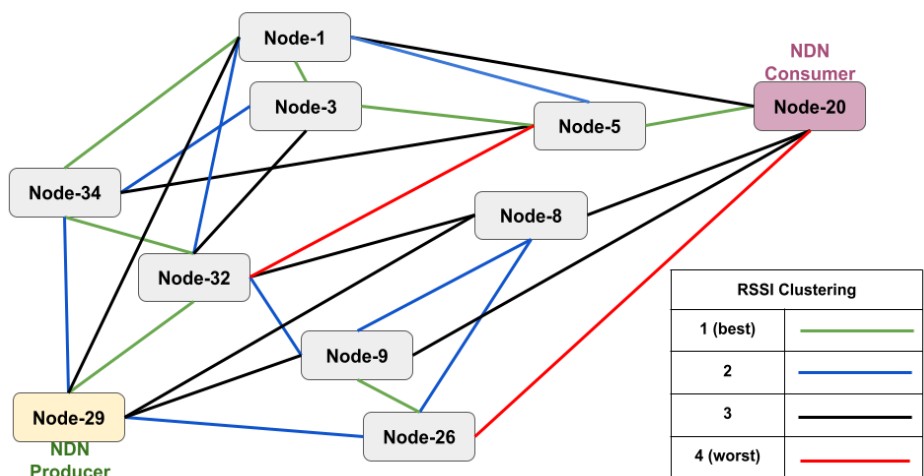

**Figure 10.** RSSI clustering results among network nodes.

The second part of this scenario includes: (i) the performance evaluation of the available NDN paths over the considered WMN and (ii) the comparison of the clustering results with the choices of our dynamic SDN-based solutions. Specifically, in this phase, we

performed 15,000 Interest-data exchanges (using the reactive process) over the topology described in Figure 5, and we recorded the BATMAN path choices (i.e., which paths have been chosen and how many times). Then, we performed 1500 Interest-data exchanges and 100 file transfers for each selected path using the NDN chunk method, and finally, for each path, we present the clustering results.

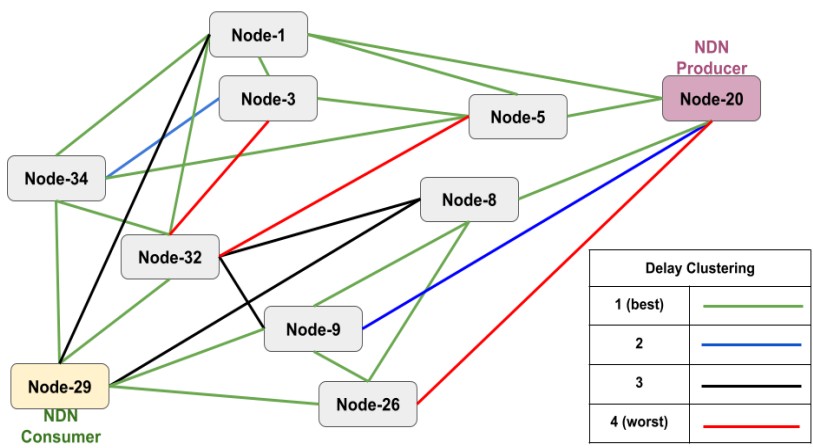

**Figure 11.** Delay clustering results among network nodes.

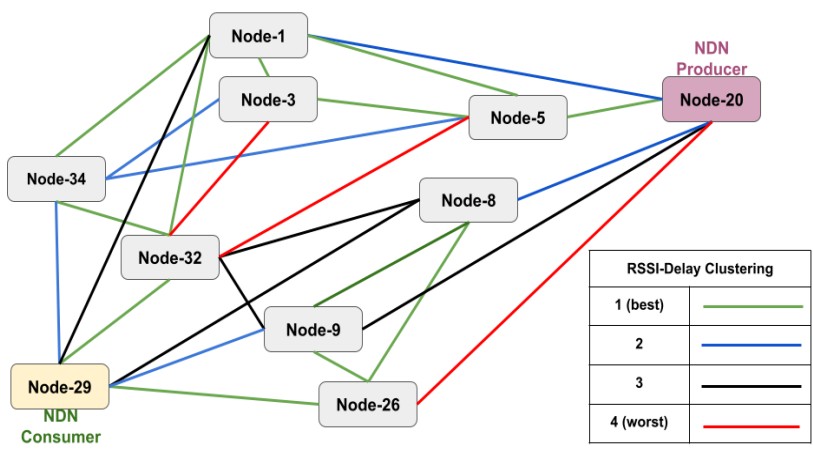

**Figure 12.** RSSI-Delay clustering results among network nodes.

Table 5 enlists: (i) the reactive's solution selected paths out of a total of 15,000 interest-data exchanges (Paths); (ii) the hops number of each path (Hops); (iii) the times each path was chosen during the experiment (reactive's solution path choices); (iv) the average delay for the Interest-Data exchanges (Interest-Data); (v) the percentage of Interest-Data exchange failures (Fails %), and (vi) the clustering results, for the metrics under consideration, i.e., RSSI, Delay, and RSSI-Delay. Here, notation C.i, $i = \{1, 2, 3, 4\}$ refers to paths sorted from best to worst. A path is characterized by its worst link, for example, C.1 exclusively includes the best links in terms of clustering and C.4 includes at least one link estimated as the worst link.

We discuss the results focusing on the Interest-Data exchange performance. In this context, Table 5 highlights that (i) the number of hops does not affect the Interest-Data performance and (ii) the reactive solution's path choices do not correspond to the best path selection of the clustering approach, for example, the path 29-8-20, No.14; (iii) reactive solution avoids unreliable paths, i.e., paths characterized by significant failures (Fails %).

Regarding the clustering outputs, one may observe that the Delay based and the RSSI-Delay based clustering successfully reveal low delay paths, while, this is not the case in the RSSI-based clustering, for example, the path with the minimum delay is characterized as C.3). Finally, all three clustering realizations provide similar results in terms of reliability.

**Table 5.** Comparison of the clustering results with the reactive process, considering the average delay and fails (%) of 1500 interest-data exchanges, over each of the available NDN paths.

| No. | Paths | Hops | Reactive Solution's Path Choices | Interest-Data Delay (msec) | Fails (%) | RSSI | Delay | RSSI-Delay |
|---|---|---|---|---|---|---|---|---|
| 1 | 29-32-1-20 | 3 | 42 | 14.82 | 3.2 | C.3 | C.1 | C.2 |
| 2 | 29-32-1-5-20 | 4 | 781 | 15.61 | 3.9 | C.2 | C.1 | C.1 |
| 3 | 29-34-1-20 | 3 | 52 | 16.79 | 5.7 | C.3 | C.1 | C.2 |
| 4 | 29-9-8-20 | 3 | 335 | 18.72 | 4.0 | C.3 | C.1 | C.2 |
| 5 | 29-32-34-1-5-20 | 5 | 8 | 19.22 | 2.3 | C.2 | C.1 | C.1 |
| 6 | 29-34-1-5-20 | 4 | 624 | 19.5 | 6.5 | C.2 | C.1 | C.2 |
| 7 | 29-26-8-20 | 3 | 359 | 21.67 | 5.1 | C.3 | C.1 | C.2 |
| 8 | 29-32-34-1-3-5-20 | 6 | 4 | 23.16 | 2.3 | C.1 | C.1 | C.1 |
| 9 | 29-32-1-3-5-20 | 5 | 40 | 26.79 | 3.9 | C.2 | C.1 | C.2 |
| 10 | 29-34-1-3-5-20 | 5 | 21 | 27.75 | 5.9 | C.2 | C.1 | C.2 |
| 11 | 29-1-5-20 | 3 | 1419 | 28.48 | 8.9 | C.3 | C.3 | C.3 |
| 12 | 29-1-20 | 2 | 208 | 33.46 | 10.9 | C.3 | C.3 | C.3 |
| 13 | 29-34-5-20 | 3 | 528 | 38.14 | 19.2 | C.3 | C.1 | C.2 |
| 14 | 29-8-20 | 2 | 5798 | 39.68 | 9.1 | C.3 | C.3 | C.3 |
| 15 | 29-32-34-5-20 | 4 | 5 | 43.27 | 23.5 | C.3 | C.1 | C.2 |
| 16 | 29-34-3-5-20 | 4 | 173 | 48.39 | 8.9 | C.2 | C.2 | C.2 |
| 17 | 29-32-34-3-5-20 | 5 | 20 | 54.68 | 8.9 | C.2 | C.2 | C.2 |
| 18 | 29-9-20 | 2 | 3758 | 54.92 | 6.2 | C.3 | C.2 | C.3 |
| 19 | 29-26-9-20 | 3 | 87 | 79.03 | 11.6 | C.3 | C.3 | C.3 |
| 20 | 29-32-8-20 | 3 | 398 | 79.95 | 22.4 | C.3 | C.3 | C.3 |
| 21 | 29-32-9-20 | 3 | 31 | 123.77 | 14.6 | C.3 | C.3 | C.3 |
| 22 | 29-26-20 | 2 | 28 | 144 | 21.1 | C.4 | C.4 | C.4 |
| - | Reactive solution (total) | 2 to 6 | 15,000 | 89.19 | 4.7 | - | - | - |

Table 6 presents the number of hops over all the available paths in the corresponding network topology, and, the total amount and the percentage of hops included in the path selection of the reactive process, over 15,000 Interest-data exchanges. As described in Table 6, the reactive solution tends to select paths with the fewest hops, e.g., the paths with two hops are selected for 65% of the total requests.

Here, we evaluate the performance of NDN over the aforementioned static NDN paths (i.e., Table 5) considering an NDN application that generates long flows. More precisely, Table 7 depicts the average delay (in secs) of transferring a file, sized 1 Mb, as NDN data segments utilizing the *ndncatchunks and ndnputchunks* tools. In this particular application, we enable NDN retransmissions and consequently, all requests are served successfully. The results in Table 7 reconfirm the conclusions of the previous use case. More specifically: (i) the RSSI-based clustering does not efficiently categorize the paths in terms of performance, since the lower delay values correspond equally to the C.2 and

C.3 categories; (ii) delay-based clustering successfully identifies links with high delay performance. Interestingly, paths are categorized as C.1, such as the path 29-32-34-5-20 (No. 16), show inefficient delay, due to the influence of retransmissions on the total delay; (iii) The RSSI-Delay clustering provides comparable outputs to that of delay clustering.

**Table 6.** Number of hops included on each reactive's solution path choice, over 1500 interest-data exchanges.

| Number of Hops | Total Amount of Reactive's Solution Choices | Percentage of Reactive's Solution Choices (%) |
|:---:|:---:|:---:|
| 2 | 9792 | 65.28 |
| 3 | 3251 | 21.67 |
| 4 | 1583 | 10.55 |
| 5 | 93 | 0.62 |
| 6 | 4 | 0.03 |

**Table 7.** Comparison of the clustering results, considering the average delay (secs) of 1MB file transfer using NDN-Chunks application.

| N. | Paths | NDN-Chunks Delay (secs) | RSSI | Delay | RSSI-Delay |
|:---:|:---:|:---:|:---:|:---:|:---:|
| 1 | 29-34-1-20 | 11.48 | C.3 | C.1 | C.2 |
| 2 | 29-32-1-20 | 11.57 | C.3 | C.1 | C.2 |
| 3 | 29-26-8-20 | 11.72 | C.3 | C.1 | C.2 |
| 4 | 29-34-1-5-20 | 12.09 | C.2 | C.1 | C.2 |
| 5 | 29-9-8-20 | 12.26 | C.3 | C.1 | C.2 |
| 6 | 29-32-1-5-20 | 12.27 | C.2 | C.1 | C.1 |
| 7 | 29-34-5-20 | 12.7 | C.3 | C.1 | C.2 |
| 8 | 29-34-1-3-5-20 | 12.95 | C.2 | C.1 | C.2 |
| 9 | 29-32-34-1-5-20 | 12.98 | C.2 | C.1 | C.1 |
| 10 | 29-34-3-5-20 | 13.03 | C.2 | C.2 | C.2 |
| 11 | 29-32-1-3-5-20 | 13.18 | C.2 | C.1 | C.2 |
| 12 | 29-32-34-1-3-5-20 | 13.82 | C.1 | C.1 | C.1 |
| 13 | 29-32-34-3-5-20 | 15.24 | C.2 | C.2 | C.2 |
| 14 | 29-1-20 | 16.89 | C.3 | C.3 | C.3 |
| 15 | 29-1-5-20 | 16.9 | C.3 | C.3 | C.3 |
| 16 | 29-32-34-5-20 | 18.74 | C.3 | C.1 | C.2 |
| 17 | 29-8-20 | 20.18 | C.3 | C.3 | C.3 |
| 18 | 29-32-8-20 | 24.82 | C.3 | C.3 | C.3 |
| 19 | 29-9-20 | 25.37 | C.3 | C.2 | C.3 |
| 20 | 29-26-9-20 | 25.59 | C.3 | C.3 | C.3 |
| 21 | 29-32-9-20 | 27.11 | C.3 | C.3 | C.3 |
| 22 | 29-26-20 | 31.36 | C.4 | C.4 | C.4 |

Tables 8–10, summarize the clustering results, over Delay, RSSI, and RSSI-Delay, by evaluating the average performance for both applications, i.e., NDN Interest-Data exchanges and 1MB file transfer using the NDN-Chunks application). More specifically, we

measure for each cluster: (i) the number of paths; (ii) the percentage of reactive's solution path choices per cluster out of the 15,000 interest-data exchanges; (iii) the average delay of the Interest-Data exchange (msec) carried out on the paths belonging to each cluster; (iv) the percentage of failed Interest-Data exchanges, and (v) the average delay of the file transfer, considering the chunk application. Results indicate that the proposed clustering solutions efficiently categorize the paths, i.e., the average values of delays (for both applications) and fails increase according to the clustering characterization.

**Table 8.** RSSI clusters average performance.

| RSSI | No. of Paths | Reactive Solution's Path Choices (%) | Interest-Data Delay (msec) | Fails (%) | NDN-Chunks Delay (secs) |
|---|---|---|---|---|---|
| C.1 | 1 | 0.03 | 23.16 | 2.33 | 13.82 |
| C.2 | 7 | 11.11 | 30.28 | 5.76 | 13.11 |
| C.3 | 13 | 86.80 | 45.59 | 11.11 | 18.10 |
| C.4 | 1 | 0.19 | 144.00 | 21.13 | 31.36 |

**Table 9.** Delay clusters average performance.

| Delay | No. of Paths | Reactive Solution's Path Choices (%) | Interest-Data Delay (msec) | Fails (%) | NDN-Chunks Delay (secs) |
|---|---|---|---|---|---|
| C.1 | 12 | 18.66 | 23.79 | 7.13 | 12.98 |
| C.2 | 3 | 26.34 | 52.66 | 8.02 | 17.88 |
| C.3 | 6 | 52.94 | 64.06 | 12.92 | 21.92 |
| C.4 | 1 | 0.19 | 144.00 | 21.1 | 31.36 |

**Table 10.** RSSI-Delay clusters average performance.

| RSSI-Delay | No. of Paths | Reactive Solution's Path Choices (%) | Interest-Data Delay (msec) | Fails (%) | NDN-Chunks Delay (secs) |
|---|---|---|---|---|---|
| C.1 | 2 | 5.29 | 19.33 | 2.84 | 13.02 |
| C.2 | 11 | 14.66 | 30.05 | 8.62 | 13.18 |
| C.3 | 7 | 77.99 | 72.91 | 13.11 | 23.53 |
| C.4 | 1 | 0.19 | 144 | 21.13 | 31.36 |

Based on the analysis of the scenario's 2 results, we can synthesize our conclusions in the following two points:

- The usage of dynamic protocols over relatively stable wireless mesh networks is not always the best solution in terms of delay, i.e., the reactive solution, which is based on BATMAN routing protocol, does not always select the "best" NDN path.
- Clustering is an efficient method for determining the "best" NDN paths over a stable WMN, considering both performance and reliability, for example, a clustering technique incorporating both RSSI and delay metrics may successfully locate the paths with low delay and interest data exchange failures.

Concluding this Section, in Table 11, we present the major advantages/disadvantages of both approaches, derived from the experimental analysis. More precisely, the overall results provided a piece of strong evidence that there is not a one-fits-all approach for the establishment of the appropriate NDN path over WMNs. More importantly, efficient rooting strategies should consider the communication conditions of the network environment under consideration.

**Table 11.** Comparison of reactive and proactive NDN path selection strategies.

| Strategy | Advantages | Disadvantages |
|---|---|---|
| **Reactive NDN path selection based on BATMAN protocol** | Reliability and fault tolerance | Control management overhead |
| | Adaptation to unstable conditions (topology rearrangements) | Need to implement a reliable channel for the control plane |
| | Rapid detection of network changes (e.g., connection failures) | High complexity, support of small-scale topologies |
| **Proactive cluster-based NDN path selection** | Routing decisions based on Delay and RSSI quality measurements | Lack of adaptation to dynamic changes |
| | Support of larger topologies | Ignores the current network state |
| | Low complexity and control overhead | Does not detect topology changes |

## 5. Conclusions and Future Work

Smart-city environments incorporate a huge amount of nodes deployed in large areas, highlighting the need for efficient multi-hop NDN communication based on appropriate NDN path selection. To address the latter, in this paper, we have discussed two alternative SDN-based NDN strategies, involving a reactive and a proactive solution, extensively evaluated over a real WMN smart-city testbed. We investigated the potential gains of each approach in terms of end-to-end NDN delay performance, considering several use cases and wireless communication conditions. The real experiments demonstrated that i) a reactive approach is preferable in communication environments with frequent network changes since the real-time centralized management increases the network overhead; ii) a proactive solution (e.g., clustering based) may successfully identify wireless routes with high performance and reliability, considering smart-city deployments with stable network conditions.

In future work, we plan to

- develops a hybrid-protocol SDN platform involving mechanisms to distinguish between smart-city regions with stable and unstable network communication conditions, deploying accordingly the appropriate NDN path selection strategy.
- extend the controller's decision-making capabilities: (a) involving additional NDN-related parameters, for example, caching information of the intermediate network nodes and (b) elaborating improvements on both (reactive/proactive) NDN path selection strategies, based on AI/ML algorithms.
- validate the proposed NDN path selection approaches over large-scale WMN, with multiple consumers and producers.
- compare experimentally our SDN-based solutions with non-SDN strategies.

**Author Contributions:** Conceptualization, S.K. and L.M.; Methodology, S.K., S.S., V.D. and L.M.; Software, S.K., S.S. and V.D.; Validation, S.K. and S.S.; Formal analysis, S.K., S.S. and V.D.; Investigation, S.K. and L.M.; Resources, S.K.; Writing—review & editing, L.M.; Supervision, S.S., L.M. and V.T.; Project administration, L.M. and V.T. All authors have read and agreed to the published version of the manuscript.

**Funding:** This work received funding from EU's H2020 research and innovation programme through the 9th open call scheme of the Fed4FIRE+ project (grant agr. num. 732638). It is also co-funded by Greece and the European Union (European Social Fund-ESF) through the Operational Programme "Human Resources Development, Education and Lifelong Learning" in the context of action "Enhancing Human Resources Research Potential by undertaking a Doctoral Research," sub-action 2: "IKY Scholarship Programme for Ph.D. candidates in Greek Universities".

**Institutional Review Board Statement:** Not applicable.

**Informed Consent Statement:** Not applicable.

**Data Availability Statement:** Not applicable.

**Conflicts of Interest:** The authors declare no conflict of interest.

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
