# Peer review of "Logically-Centralized SDN-Based NDN Strategies for Wireless Mesh Smart-City Networks†"

_futureinternet, doi:10.3390/fi15010019_

Round 1

Reviewer 1 Report

In my opinion, the article raises an important and interesting issue regarding the management of networks connecting IoT elements in urban infrastructure. The article is very interesting and prompted me to take a closer look at some of the publications cited in the paper. A strong point of the article is the presentation of results implemented on a real test bed. In scientific journals, we can find a great deal of theoretical work on SDN networks, which, however, do not include descriptions of the practical aspects of their implementation. In contrast to these works, the authors of the reviewed article presented results from the practical implementation of their solution. 

I would like the authors to clearly indicate the areas in which WMN solutions can be applied and have an advantage over existing or planned to be implemented 5G and 6G systems.

In conclusion, I wanted to recommend the article for publication. 

Reviewer 2 Report

This paper compares two strategies (reactive and proactive) for deploying NDN over SDN-based Wireless Mesh Networks, with smart city applications in mind. The paper is interesting and definitely has some merits. The bulk of the paper focuses on presenting results to decide whether it is more suitable to find the best NDN overlay path dynamically (“reactive”), or to set up the path in advance using partitional clustering. The problem is that the paper does not present a solid conclusion, or at least does not convincingly analyze the results to indicate the superior performance of one approach over the other.

There are some questions regarding the overall validity of the work. The main concern about using SDN-NDN over WMN, as authors have pointed out, is the overhead and scalability. However the paper does not seem to consider these two factors in performance analysis. If as indicated in Figure 2 each new content request must be sent to the SDN controller, this by itself would create a major bottleneck in terms of resources and performance at the controller. Also the authors assume a separate and totally reliable control plane network for communication between NDN nodes and the SDN controller. This assumption may not be valid in a WMN.

Most references (and in particular [16] and [20]) have mentioned efficient in-network caching as a major benefit of SDN. However, it seems this paper only considers end-to-end delivery with no in-network caching.  This is a major flaw in the work. At minimum, the authors should explain how their approach would perform when in-network caching is considered.

Without in-network caching, it becomes hard to justify the use of SDN in this scenario. According to Figure 6, about 90% of the average total delay is due to communication with the SDN controller. How would the NDN network perform if it was implemented in a distributed manner rather than a logically centralized SDN?

Apart from these concerns, the biggest problem with the paper is in its presentation of results data. Most of Tables 2-10 are quite illegible to anyone who is not familiar with authors’ intention and terminology.  Table values are not clearly defined and units are not properly identified. In many tables, it seems comma (,) has been used instead of period (.) to identify a decimal point, which was extremely confusing. Table 3 and 4 present the value of “failures” according to their caption; however, it is not clear if this represent a failure percentage or an absolute value, and if it is an absolute value, how does it compare to the total value. For instance, is a failure of 115 a high failure number or an acceptable failure number? This value by itself has no meaning. Table 6 presents “BATMAN multihop choices” in which the number of choices and a percentage has been given, but it is not clear what that percentage represents. Table 7-10 seem to include the main comparative results, however they are also the most confusing and illegible. What is chunks which has a unit of seconds? What is the “Batman choices %” in those tables?

The final conclusion is also not strongly supported by the results. While the paper claims that a combined RSSI-Delay metric provides superior delay and reliability performance, the difference in results in Tables 8-10  is fairly marginal. The paper must provide a comparative table that clearly compares the reactive and proactive methods side by side and demonstrates the superiority of the selected method in terms of the chosen performance metric (Delay and overhead). It would be even more enhanced if it also provides a comparison with a non-SDN distributed NDN strategy too. After all, the use of SDN in this scenario must be justified in the first place, and neither this paper nor the authors’ earlier conference paper have provided a solid justification.      

Round 2

Reviewer 2 Report

Thanks to the authors for taking the prior round of reviews into account and for improving the paper's content, style and presentation. While in my opinion the limited performance evaluations and the lack of comparison with distributed solutions somewhat diminishes the novelty and significance of the paper, I believe it can be published in the current form.

One short note regarding the use of separate channels for SDN control and data planes: While this approach may work for wired SDN, in wireless domain it does not solve the reliability problem because anything that causes the failure of the wireless data link, would likely fails the wireless control link as well.  having said that, the assumption of separate channels is commonly made in SDN research and the authors could not be faulted for using it.